# Electron tomography visualization of HIV-1 fusion with target cells using fusion inhibitors to trap the pre-hairpin intermediate

**Mark S Ladinsky[1], Priyanthi NP Gnanapragasam[1], Zhi Yang[1], Anthony P West[1], Michael S Kay[2], Pamela J Bjorkman[1]***

[1]Division of Biology and Biological Engineering, California Institute of Technology, Pasadena, United States; [2]Department of Biochemistry, University of Utah School of Medicine, Salt Lake City, United States

**Abstract** Fusion of HIV-1 with the membrane of its target cell, an obligate first step in virus infectivity, is mediated by binding of the viral envelope (Env) spike protein to its receptors, CD4 and CCR5/CXCR4, on the cell surface. The process of viral fusion appears to be fast compared with viral egress and has not been visualized by EM. To capture fusion events, the process must be curtailed by trapping Env-receptor binding at an intermediate stage. We have used fusion inhibitors to trap HIV-1 virions attached to target cells by Envs in an extended pre-hairpin intermediate state. Electron tomography revealed HIV-1 virions bound to TZM-bl cells by 2–4 narrow spokes, with slightly more spokes present when evaluated with mutant virions that lacked the Env cytoplasmic tail. These results represent the first direct visualization of the hypothesized pre-hairpin intermediate of HIV-1 Env and improve our understanding of Env-mediated HIV-1 fusion and infection of host cells.

***For correspondence:**
bjorkman@caltech.edu

## Introduction

The first step of HIV-1 entry into a host target cell, fusion between the viral and target cell membranes, is mediated by the viral envelope spike protein (Env). HIV-1 Env is a trimeric glycoprotein comprising three gp120 subunits that contain host receptorbinding sites and three gp41 subunits that include the fusion peptide and membrane-spanning regions. Binding of the primary receptor CD4 to gp120 triggers conformational changes that expose a binding site for co-receptor (CCR5 or CXCR4). Coreceptor binding results in further conformational changes within gp41 that promote release of the hydrophobic fusion peptide, its insertion into the host cell membrane, and subsequent fusion of the host cell and viral membrane bilayers (*Harrison, 2015*).

Structural studies relevant to understanding Env-mediated membrane fusion include X-ray and single-particle cryo-EM structures of soluble native-like Env trimers in the closed (pre-fusion) conformation (*Ward and Wilson, 2017*), CD4-bound open trimers in which the co-receptor binding site on the third hypervariable loop (V3) of gp120 is exposed by V1V2 loop rearrangement (*Ozorowski et al., 2017*; *Wang et al., 2018*; *Wang et al., 2016*; *Yang et al., 2019*), a gp120 monomeric core-CD4-CCR5 complex (*Shaik et al., 2019*), and a post-fusion gp41 six-helical bundle formed by an α-helical trimeric coiled coil from the gp41 N-trimer region surrounded by three helices from the C-peptide region (*Chan et al., 1997*; *Weissenhorn et al., 1997*; *Figure 1a*). Prior to membrane fusion and formation of the post-fusion gp41 helical bundle, the viral and host cell membranes are hypothesized to be linked by an extended pre-hairpin intermediate in which insertion of the gp41 fusion peptide into the host cell membrane exposes the N-trimer (HR1) region of gp41

(*Chan and Kim, 1998*). Formation of the six-helical bundle and subsequent fusion can be inhibited by targeting the N-trimer region with C-peptide-based inhibitors; for example the fusion inhibitor T20 (enfuvirtide [Fuzeon]) (*Kilgore et al., 2003*), T1249, a more potent derivative of T20 (*Eron et al., 2004*), and a highly potent trimeric D-peptide (CPT31) (*Redman et al., 2018*), or with anti-gp41 antibodies such as D5 (*Miller et al., 2005*; *Figure 1a*).

Visualizing the pre-hairpin intermediate that joins the host and viral membranes has not been straightforward. Despite 3-D imaging by electron tomography (ET) of HIV-1 infection of cultured cells (*Carlson et al., 2008*; *Carlson et al., 2010*; *Do et al., 2014*; *Earl et al., 2013*) and tissues (*Kieffer et al., 2017a*; *Kieffer et al., 2017b*; *Ladinsky et al., 2019*; *Ladinsky et al., 2014*), viruses caught in the act of fusion have not been unambiguously found. In our ET imaging of HIV-1–infected humanized mouse tissues, we have identified hundreds of budding virions at various stages of egress and thousands of free mature and immature virions (*Kieffer et al., 2017a*; *Kieffer et al., 2017b*; *Ladinsky et al., 2019*; *Ladinsky et al., 2014*), but not a single example of a virus attached to a host cell via a pre-hairpin intermediate or in the process of fusing its membrane with the target cell membrane. The absence of observed viral fusion events might be explained if fusion is a fast process compared with viral budding; thus when cells or tissues are immobilized for EM or ET, the relatively slow process of viral budding would be more easily captured compared with the faster process of fusion. We assume that fusion could theoretically be observed if a virus were caught at exactly the right time, but this might require examining thousands or millions of images.

Here, we report visualizing the pre-hairpin intermediate by ET after treatment of HIV-1–exposed target cells with inhibitors of six-helix bundle formation that bind the N-trimer region of gp41 that is exposed during the fusion process. Using optimally preserved samples for ET with a nominal resolution ~7 nm, we found >100 examples of HIV-1 virions linked to TZM-bl target cells by 2–4 narrow rods of density (spokes) in inhibitor-treated samples, but none in untreated or control-treated samples. The approximate dimensions of the majority of the spokes (*Ausubel et al., 1989*) matched models of gp41-only pre-hairpin intermediates in which the Env gp120 subunit had been shed. The average number of observed spokes connecting a virion to a target cell increased slightly when using a virus containing an Env with a cytoplasmic tail deletion, suggesting that the increased lateral mobility of cytoplasmic tail-deleted Envs in the viral membrane (*Crooks et al., 2008*; *Roy et al., 2013*; *Pezeshkian et al., 2019*) allowed more Envs to join the interaction with the target cell. We discuss the implications of these studies for understanding HIV-1 Env-mediated membrane fusion and how these results differ from a previous ET study of the 'entry claw' that is formed upon HIV-1 or SIV interactions with target cells (*Sougrat et al., 2007*).

## Results

### Experimental design

A previous study used ET to visualize HIV-1 and SIV virions in contact with target cells after promoting a temperature-arrested state (*Mkrtchyan et al., 2005*) in which viruses can remain attached to cells prior to fusion (*Sougrat et al., 2007*). For that study, target cells were incubated with virus at 4°C to allow binding but not fusion, warmed to 37°C, and then fixed after incubations ranging from 15 min to 3 hr (*Sougrat et al., 2007*). At all time points after warming, viruses were found attached to target cells by a cluster of 5–7 'rods,' each ~100 Å long and ~100 Å wide. The fact that the attachment structure was not found when the viruses and target cells were incubated in the presence of C34, a gp41 N-trimer–targeting C-peptide inhibitor related to T20 (*Sougrat et al., 2007*), suggests that the rod structure that was trapped during the temperature-arrested state did not involve the pre-hairpin intermediate.

We hypothesized that addition of an HIV-1 fusion inhibitor that binds to the exposed gp41 N-trimer after host cell receptor and coreceptor binding would slow or stop virus-host cell membrane fusion such that we could visualize pre-hairpin intermediate structures by ET (*Figure 1b*). We characterized three fusion inhibitors of different sizes and potencies that target the exposed gp41 N-trimer region of the pre-hairpin intermediate for attempts to visualize the pre-hairpin intermediate: T1249-Fc, a C-peptide–based inhibitor that we linked to human Fc (MW = 65 kDa), D5 IgG (MW = 150 kDa) (*Miller et al., 2005*), and CPT31, a high-affinity D-peptide inhibitor linked to cholesterol (MW = 9 kDa) (*Redman et al., 2018*; *Welch et al., 2010*; *Figure 1a*; *Figure 1—figure supplement*

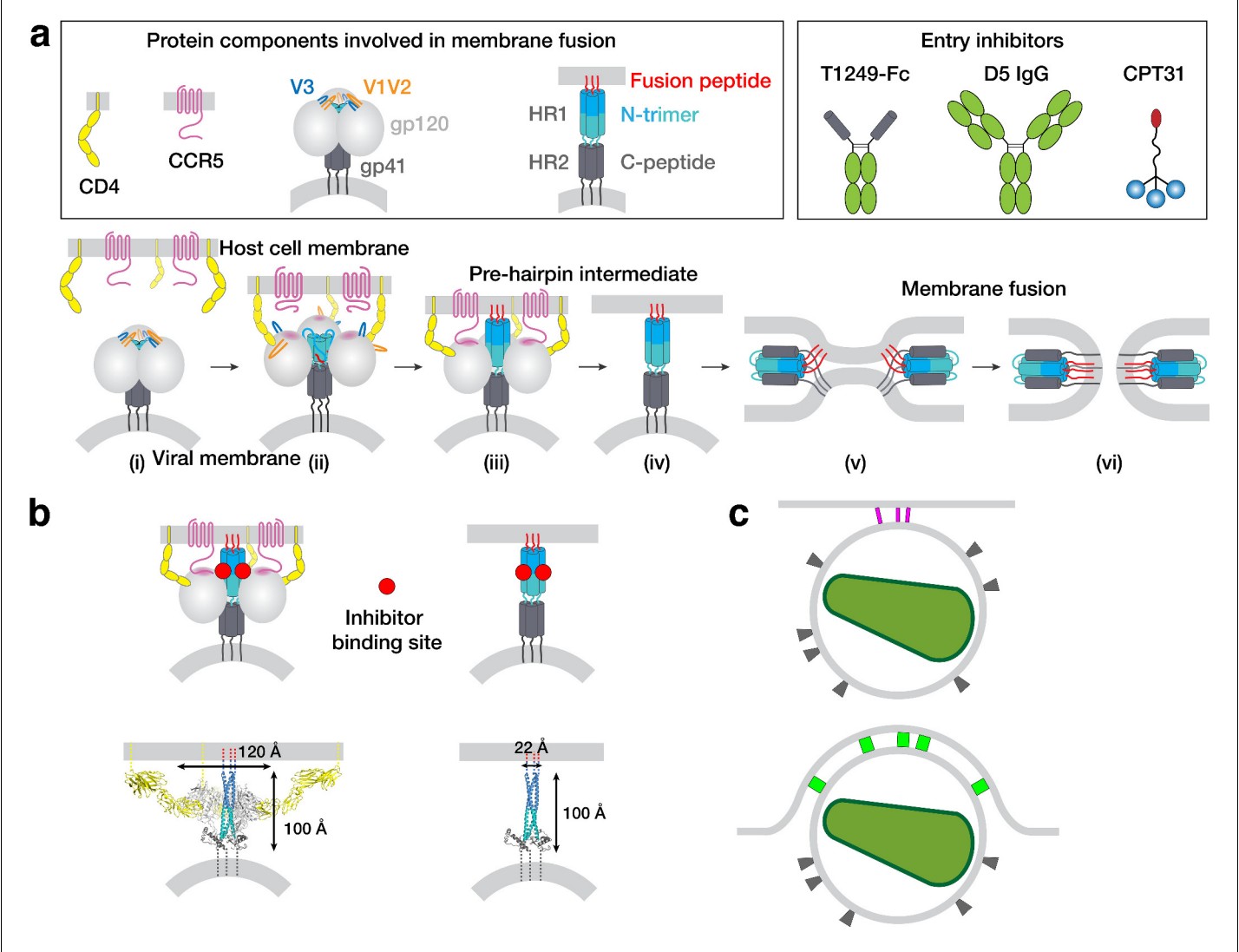

**Figure 1.** HIV-1 Env-mediated fusion between viral and host cell membranes. (**a**) Top: Schematics of host receptors, HIV-1 Env trimer, pre-hairpin intermediate, and fusion inhibitors. Bottom: steps in fusion: (i) Closed, prefusion structure of HIV-1 Env trimer in which the V1V2 loops (orange) occlude the coreceptor-binding site on V3 (blue) (e.g., PDB code 5CEZ). The Env trimer is embedded in the viral membrane, while the host receptor (CD4) and coreceptor (CCR5) are embedded in the target cell membrane. (ii) CD4-bound open HIV-1 Env trimer in which V1V2 loops have been displaced to expose the coreceptor-binding site on V3 (e.g. PDB 6U0L). (iii) Hypothetical CD4- and CCR5-bound open Env trimer with rearrangements of gp41 N-trimer/HR1 to form a pre-hairpin intermediate structure that is linked to the target cell membrane by the gp41 fusion peptide (red). (iv) Hypothetical pre-hairpin intermediate formed by gp41 trimer after shedding of gp120s. (v–vi) Formation of the post-fusion gp41 six-helical bundle (e.g. PDB 1GZL) that juxtaposes the host cell and viral membranes (step v) for subsequent membrane fusion (step vi). (**b**) Approximate binding sites (red circles) for fusion inhibitors shown on schematics of steps iii and iv (panel a). Entry inhibitor binding sites might be partially sterically occluded for binding to the T1249-Fc or D5 fusion inhibitors. Schematics shown above as models are from PDB codes 6U0L and 1AIK with approximate dimensions indicated. (**c**) Schematic illustrating why fewer HIV-1 Envs might be involved in attaching to a target cell when the attachment site is flat versus a concave surface. Top: attachment site (described here) formed during a 37°C incubation of virions, target cells, and a fusion inhibitor. Bottom: attachment site (described in *Sougrat et al., 2007*) formed in the absence of a fusion inhibitor when virions and target cells were incubated in a temperature jump protocol (4 °C incubation followed by warming to 37°C).

The online version of this article includes the following figure supplement(s) for figure 1:

**Figure supplement 1.** Characterization of fusion inhibitors and viral infectivity.

1a). We measured their neutralization potencies using in vitro HIV-1 pseudovirus neutralization assays (*Montefiori, 2009*) against the SC4226618 and 6535 viral strains. We found potencies ranging from 50% inhibitory concentration ($IC_{50}$) values of ~0.13 ng/mL for CPT31 to ≥40 µg/mL for D5

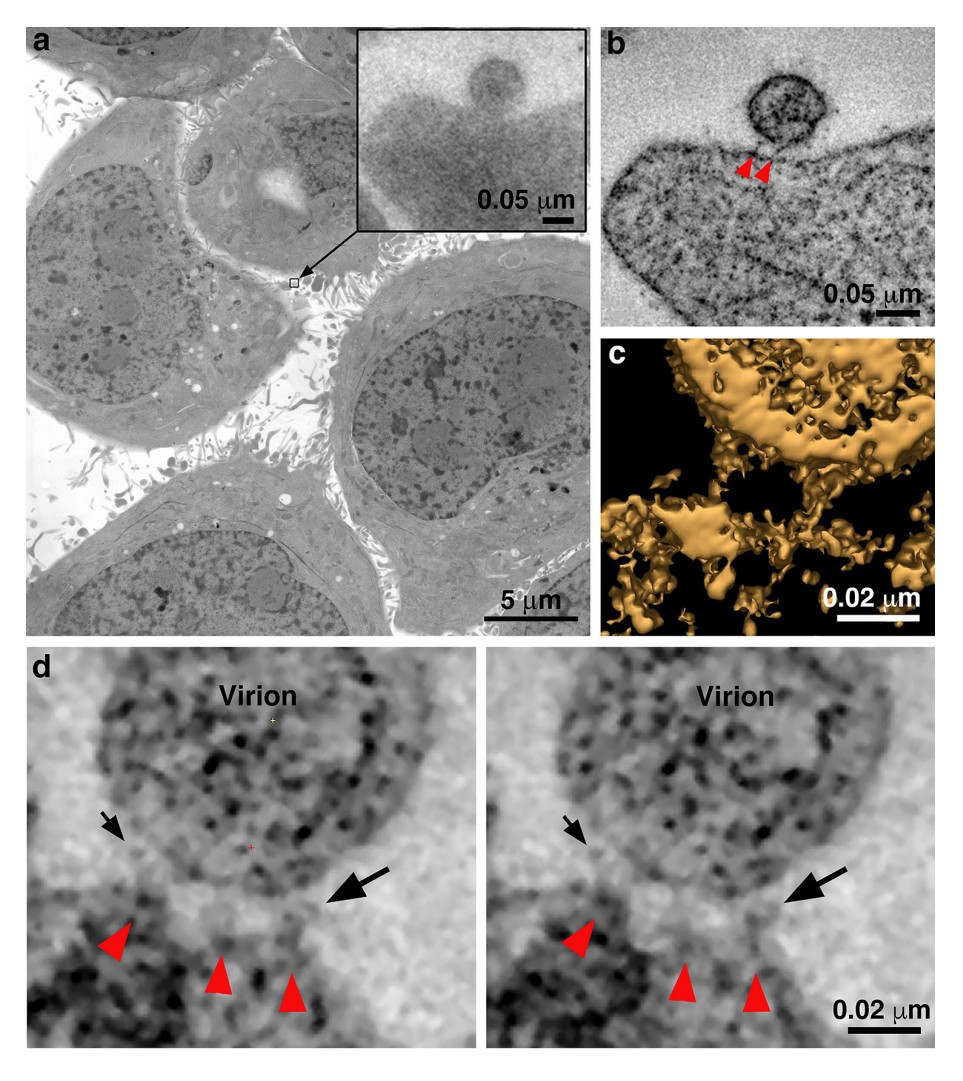

**Figure 2.** Identification of attached HIV-1 virions. (**a**) Montaged projection overview of a field of cultured TZM-bl cells from a 400 nm section. Note extensive blebbing and surface projections that are typical of the cell type. Inset: Projection detail of a HIV-1 virion adjacent to TZM-bl cell surface. (**b**) Slice (5.6 nm) from a tomographic reconstruction of the virion shown in the inset of panel a (from a dataset collected with the T1249-Fc inhibitor). The bullet-shaped core identifies the particle as mature HIV-1 (see also *Figure 3—figure supplement 1*). Two pre-hairpin intermediate 'spokes' (red arrowheads) attach the virion to the cell surface. (**c**) 3-D isosurface rendering of the spokes shown in panel b. (**d**) Examples of extra densities observed in some data sets collected using the D5 IgG inhibitor. These appear as 'hook-like' structures projecting from the sides of spokes, adjacent to the virion surface, which are visible in two sequential tomographic slices (small and large black arrows). Extra densities may represent portions of D5 IgGs attached to the prehairpin intermediate. Similar densities were not seen in experiments with the T1249-Fc or CPT31 inhibitors.

The online version of this article includes the following figure supplement(s) for figure 2:

**Figure supplement 1.** Confirmation of pseudovirions in tomograms and experimental controls.

IgG (*Figure 1—figure supplement 1a*). T1249-Fc exhibited intermediate potencies (IC$_{50}$s = 0.99 µg/mL; 17 µg/mL) (*Figure 1—figure supplement 1a*), higher than IC$_{50}$s measured for T1249 peptide alone, consistent with limited steric accessibility resulting in decreased potencies for larger fusion inhibitors (*Hamburger et al., 2005*).

Incubating with a fusion inhibitor at 37°C obviated the need for a 4°C incubation of virus and target cells, which we reasoned was desirable since low temperatures alter membrane fluidity

(*Avery et al., 1995*; *Simons and Vaz, 2004*; *Quinn, 1988*), which could affect one or more steps in membrane fusion. Since target cells for HIV-1 are several microns in height, much thicker than the 0.5–1 µm limit for cryo-ET (*Beck and Baumeister, 2016*), we used stained, plastic-embedded samples that could be cut into 300–400 nm sections using a microtome, and then examined the samples in 3-D using ET. Although ET of stained, plastic-embedded sections results in lower resolution than cryo-ET, the minimal effects of radiation damage in plastic sections (*Glaeser, 2016*) was an advantage for locating rare attached virions. Thus, more cells could be assayed in plastic sections than in samples prepared by cryo-ET methods (e.g. by examining thin leading edges of cells or using focused-ion-beam milling [*Villa et al., 2013*] to prepare a sufficiently thin sample), therefore allowing for statistically significant observations of virion attachment events. We prepared samples by light fixation followed by high-pressure freezing/freeze substitution fixation (HPF-FSF) instead of the traditional chemical fixation protocol used previously (*Sougrat et al., 2007*) because HPF vitrifies cells at ~10,000°/s, stopping all cellular movement within ms and allowing optimal preservation of ultrastructural features (*Kellenberger, 1991*; *McIntosh et al., 2005*; *Sartori et al., 1993*; *Dahl and Staehelin, 1989*). By contrast, chemical fixation immobilizes elements in the cell at different rates, and movement and rearrangement of transmembrane proteins may continue even in the presence of aldehyde fixatives (*Brock et al., 1999*; *Stanly et al., 2016*; *Tanaka et al., 2010*). Following HPF-FSF, samples were plastic embedded and stained with uranyl acetate and lead citrate as described in our previous ET studies of HIV-1 in infected tissues (*Kieffer et al., 2017b*; *Ladinsky et al., 2019*; *Ladinsky et al., 2014*). Since biosafety requirements for the current study necessitated the use of HIV-1 pseudoviruses instead of infectious HIV-1, we verified that the ultrastructure of HIV-1 pseudoviruses, including approximate numbers and dimensions of Env trimer spikes and the presence of collapsed (in mature virions) versus C-shaped (in immature virions) cores (*Carlson et al., 2008*; *Benjamin et al., 2005*; *Ganser, 1999*; *Wright et al., 2007*), was preserved during the fixation, embedding, and staining procedures (*Figure 2—figure supplement 1a*) consistent with our previous publications involving ET of infectious HIV-1 in tissue samples (*Kieffer et al., 2017b*; *Ladinsky et al., 2019*; *Ladinsky et al., 2014*). These results are also consistent with previous direct comparisons of tomograms of stained and plastic-embedded versus unstained and cryopreserved SIV virions (*Sougrat et al., 2007*).

We conducted ET experiments by first incubating TZM-bl cells, a HeLa cell line that stably expresses high levels of human CD4 and coreceptors CCR5 and CXCR4 (*Platt et al., 1998*), with 130 µg/mL of inhibitor (either T1249-Fc, D5, or CPT31) and ~5000 $TCID_{50}$/mL of HIV-1 pseudovirus at 37 °C for 2, 4, or 48 hr, followed by HPF, FSF, plastic embedding, sectioning, and visualization by ET. In order to verify that results were not dependent upon a particular viral strain, we used pseudoviruses derived from two primary isolate HIV-1 strains: SC4226618 (Tier 2) and 6535 (Tier 1B) (*Li et al., 2005*), chosen for their sensitivity to the fusion inhibitors and because we had both wild-type and Env cytoplasmic tail-deleted forms of the 6535 pseudovirus (*Figure 1—figure supplement 1a*). TZM-bl cells are contaminated with ecotropic murine leukemia virus (*Takeuchi et al., 2008*), which does not affect their use for HIV-1 in vitro neutralization assays (*Platt et al., 2009*). In our surveys of TZM-bl cells incubated in the presence or absence of fusion inhibitors, we occasionally observed budding MLV virions (*Figure 2—figure supplement 1b*). As MLV serves as a control for non-specific inhibition in TZM-bl–based HIV-1 in vitro neutralization assays (*Montefiori, 2005*), the fusion inhibitors used in our experiments are known to have no effect on MLV fusion, thus contaminating MLV virions were not captured during fusion in our experiments.

To identify attached virions by EM (*Figure 2*; *Figure 2—figure supplement 1*), the peripheries of TZM-bl cells were scanned to locate roughly spherical objects with diameters ~ 100 nm that were near a cell surface. Regions of interest were then examined at higher magnification and at tilts of 0°, 35° and −35° to verify that the objects were spherical, as expected for an HIV-1 virion (*Figure 2—figure supplement 1c*). Potential virions were then observed through a defocus series to detect core structures found inside authentic virions: that is a bullet-shaped core in mature HIV-1 and a C-shaped core in immature HIV-1 (*Benjamin et al., 2005*; *Wright et al., 2007*). Once verified as a virion, tilt series for 3-D reconstructions were collected. Control experiments in which pseudovirus and TZM-bl cells were incubated without inhibitor, with an irrelevant IgG, or with a low concentration of inhibitor, were prepared and analyzed in the same way (*Figure 2—figure supplement 1d*).

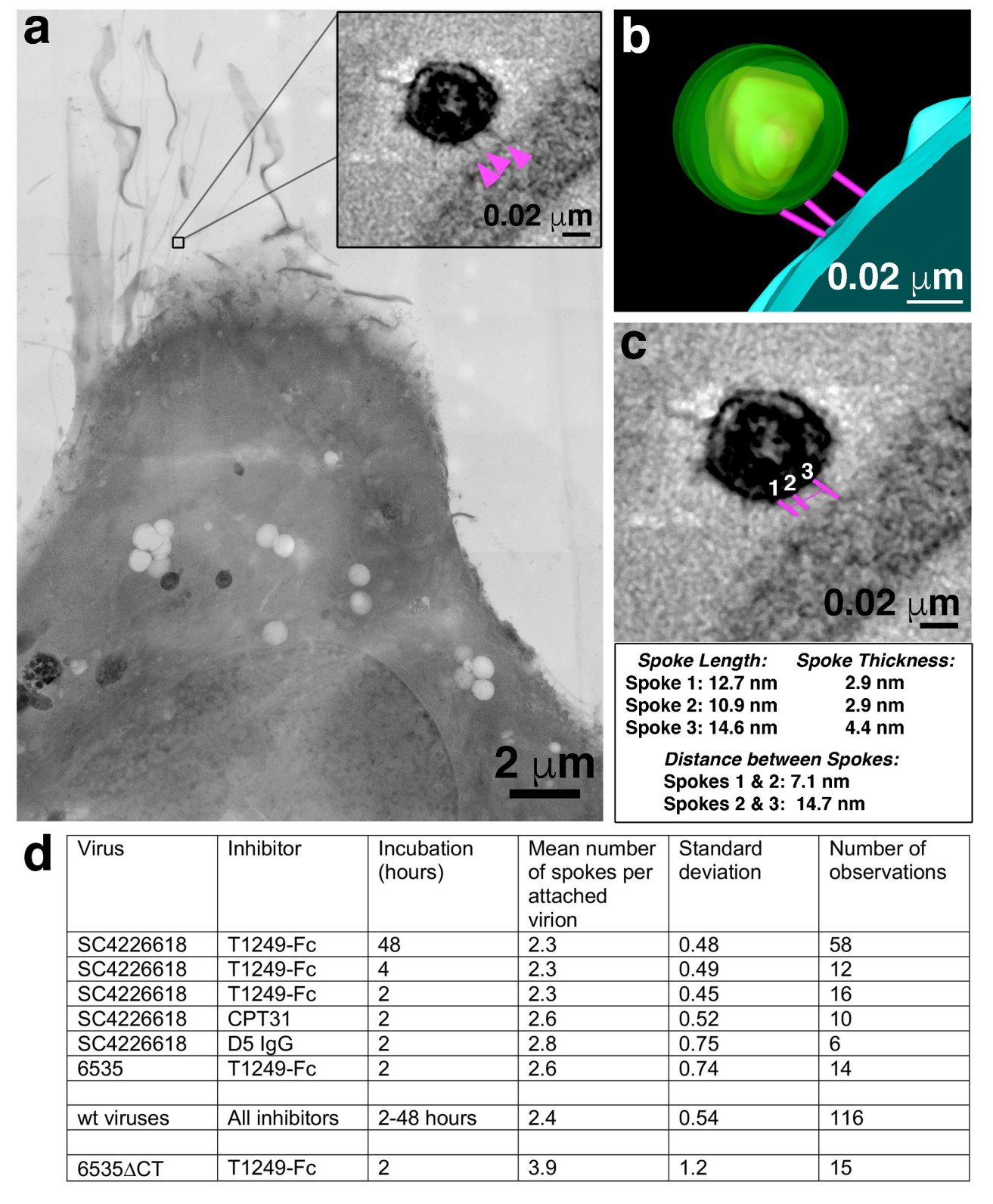

**Figure 3.** Characteristics of virions attached to target cells in the presence of a fusion inhibitor. (a) 2-D projection image of TZM-bl cell incubated with SC4226618 pseudovirions in the presence of the T1249-Fc fusion inhibitor for 2 hr at 37˚C. Inset shows a tomographic slice of the attached virion indicated by a box with attachment spokes indicated by magenta arrowheads. (b) 3-D model from tomogram of attachment site shown in panel a inset. (c) Tomographic slice of attached virion from panel b with approximate measurements of spoke length, width, and inter-spoke distances. (d) Summary

*Figure 3 continued on next page*

*Figure 3 continued*

of mean, standard deviation, and number of observations for spokes at attachment sites under different experimental conditions. See also *Figure 3—source data 1*.

The online version of this article includes the following source data and figure supplement(s) for figure 3:

**Source data 1.** Quantification of prehairpin intermediate spokes.

**Figure supplement 1.** Gallery of attachment sites formed using different fusion inhibitors and different incubation times.

## Entry inhibitor-treated virions are attached to target cells by 2–3 narrow spokes

In surveys of TZM-bl cells treated with either SC4226618 or 6535 pseudovirus and 130 µg/mL of any of the three fusion inhibitors, we found virions that were attached to the surface of a TZM-bl cell by several (usually 2 or 3) narrow densities (*Figures 2* and *3d*; *Figure 3—figure supplement 1*; *Figure 3—source data 1*). To verify that the attached virions resulted from treatment with a fusion inhibitor, we analyzed control experiments in which we incubated TZM-bl cells and pseudovirus with either no inhibitor, with an irrelevant Fc-containing protein (Z004, an anti-Zika virus IgG [*Robbiani et al., 2017*]), or with the T1249-Fc inhibitor at a concentration equivalent to 0.01x of its neutralization potency (i.e. its $IC_{50}$ value) (*Figure 1—figure supplement 1a*). In examinations of >100 cells, we found no attached virions and also very few virions that were within a distance that could accommodate attachment. We used ET to examine the few cases (<20) in which we found virions adjacent to cells, which confirmed that none of the virions were attached to a host cell membrane (*Figure 2—figure supplement 1d*).

The approximate dimensions of the densities found for virions attached to TZM-bl cells in the presence of a fusion inhibitor were on average 15.6 ± 2.8 nm in length and 3.9 nm ±0.8 nm in thickness (n = 20) (*Figure 3a–c*), thus we refer to the densities as 'spokes' to distinguish them from the wider densities described as 'rods' in the previous ET study of virions attached to cells (*Sougrat et al., 2007*). Interestingly, the spokes in our experiments, although imaged at relatively low resolution, resembled fusion intermediates observed in cryo-ET studies of hemagglutinin-mediated fusion of influenza virions with receptor-containing liposomes (*Calder and Rosenthal, 2016*) and also low pH-induced structural intermediates of hemagglutinin deduced from single-particle cryo-EM reconstructions (*Benton et al., 2020*). In some of our samples incubated with the largest inhibitor (D5 IgG), we occasionally found densities adjacent to one or more spokes that could represent the bound IgG (*Figure 2d*). We found virions attached with spokes to 10–30% of TZM-bl cells that were examined, usually located on the thin leading edges of cells. Most cells showed only one attached virion per 400 nm section, but occasional cells exhibited 3–5 attached virions. The attachment sites were generally flat, as opposed to the target cell exhibiting a concave surface corresponding to the circumference of the virion as previously described (*Sougrat et al., 2007*), with distances of ~7 nm to ~15 nm between spokes. The majority of attached virions were mature, as identified by their bullet-shaped cores (*Figures 2–4*), but a minor subset of attached virions (~5%) were immature. *Figure 2* shows overview images and 2-D tomographic slices from 3-D reconstructions of fusion inhibitor-treated virions attached to target cells; see also a gallery of examples in *Figure 3—figure supplement 1*. Attachment densities were sometimes located in different planes so they are not always visible in any given 2-D tomographic slice, thus we identified spokes in 3-D as shown in *Video 1*.

Despite using inhibitors of different sizes and at concentrations varying from 3.2- to $10^6$-fold above their $IC_{50}$ values for neutralization of the SC4226618 pseudovirus (*Figure 1—figure supplement 1a*), we found no systematic differences in the numbers of attached virions per cell, their locations on TZM-bl cells, or the numbers and dimensions of spokes per attached virion. In addition, no systematic differences were observed as a function of which fusion inhibitor or pseudovirus was included in the incubation or the length of the 37°C incubation (*Figure 3d*; *Figure 3—source data 1*). Although a 48 hr incubation at 37°C should result in a substantial loss of virus infectivity, the T1249-Fc incubation for 48 hr condition yielded an equivalent number of attached virions (*Figure 3d*) and similar spoke structures as the 2- and 4 hr incubations (*Figure 3—figure supplement 1*). This finding is rationalized by calculations and infectivity measurements showing that the

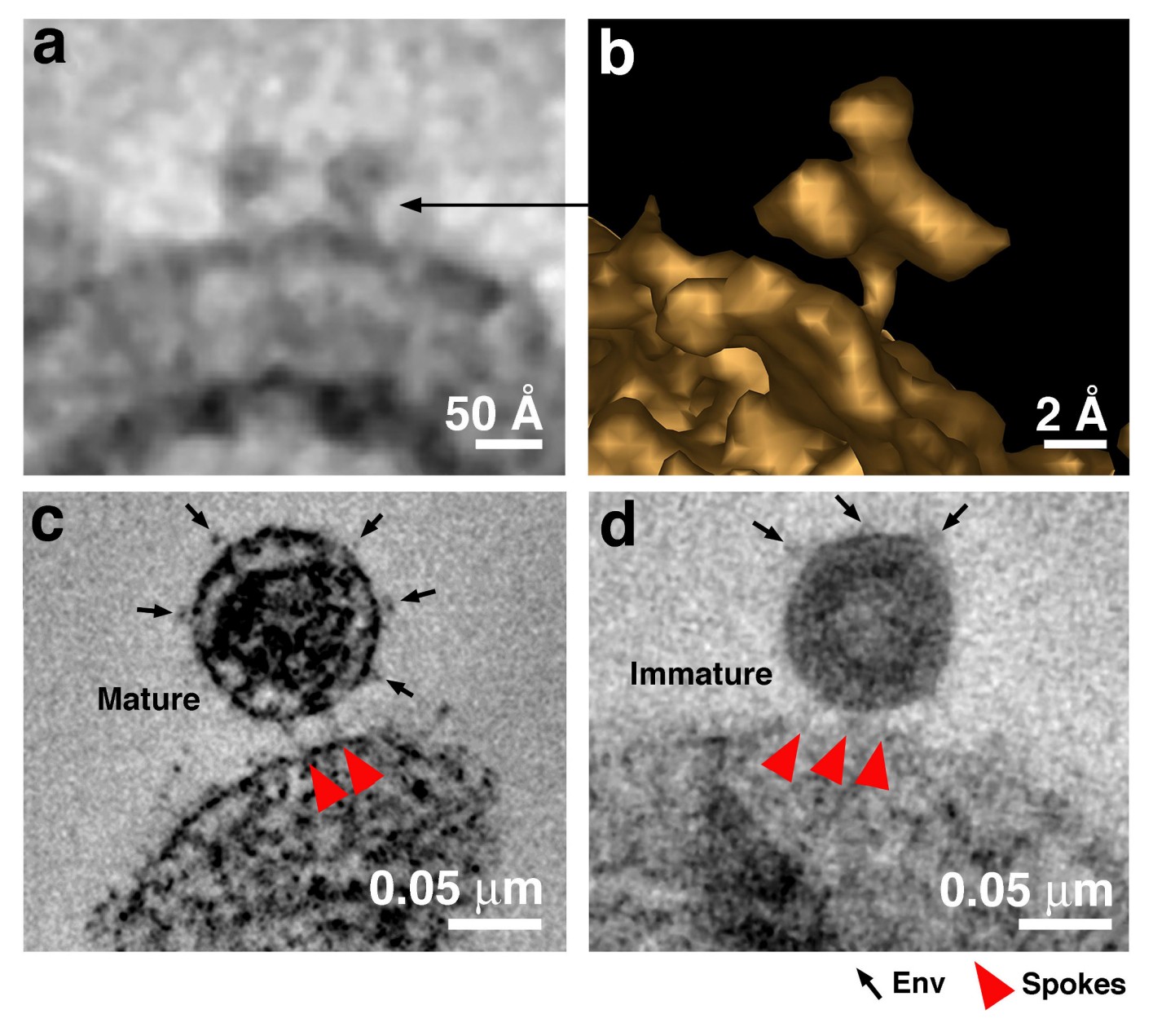

**Figure 4.** Free Env trimers can be visualized on attached virions. (**a**) Example of densities observed for free HIV-1 Env trimers in a tomographic slice. (**b**) 3-D isosurface rendering of an individual free Env trimer. (**c–d**) Examples of attached mature (panel c) and immature (panel d) virions with free Env spikes distant from the attachment site indicated by arrows. Note that free Env trimer densities and spokes at an attachment site were only rarely optimally visualized in a single tomographic slice, thus Env trimers and spokes were identified from 3-D tomograms rather than 2-D tomographic slices.

48 hr incubation conditions contained several million infectious virions (*Figure 2—figure supplement 1b*), enough to account for the observed attached virions.

## A cytoplasmic tail deletion virus forms attachment structures with more spokes

The finding of only 2–3 spokes per attached virion implies that not all of the ~14 Envs per HIV-1 virion (*Zhu et al., 2003*; *Zhu et al., 2006*; *Liu et al., 2008*; *Chertova et al., 2002*; *Layne et al., 1992*) are involved in the fusion process. Indeed, Env trimers that did not participate in attachment were sometimes observed in tomographic slices (*Figure 4*). We hypothesized that more Envs might

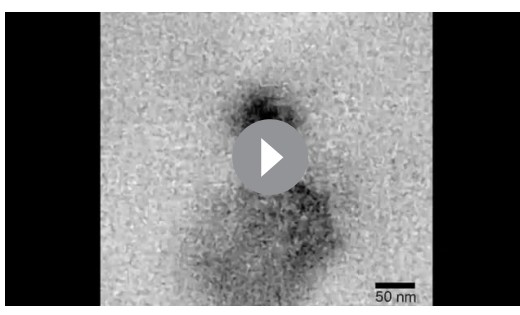

**Video 1.** Tomographic reconstruction of a mature HIV-1 pseudovirus attached to a TZM-bl cell surface by two narrow spokes. In this experiment, the T1249-Fc inhibitor was incubated with cells and SC4226618 virus at 37 ˚C for 2 hr. The movie presents the full volume of a 3-D reconstruction, advances at 1-pixel (0.5 nm) increments, and then pauses briefly to indicate the spokes (red arrowheads). The cone-shaped viral core is distinguished within the virion, as are Env trimers on the virion surface. Scale bar = 0.05 µm.
https://elifesciences.org/articles/58411#video1

join in the attachment site if we used a virus in which Envs could exhibit increased lateral mobility in the viral membrane to allow Env diffusion to the attachment site. Increased Env mobility in the viral membrane has been described in viruses with an Env cytoplasmic tail deletion that eliminates interactions with the viral matrix protein (*Crooks et al., 2008*; *Roy et al., 2013*; *Pezeshkian et al., 2019*). To evaluate potential effects of increased lateral mobility, we used a pseudovirus containing an Env with a cytoplasmic tail deletion (6535-ΔCT) and compared its attachment sites with target cells when incubated with the T1249-Fc fusion inhibitor to those of wild-type 6535 and SC4226618 pseudoviruses when incubated under the same conditions (*Figure 5*).

We first confirmed that attachment sites formed by wild-type 6535 and SC4226618 pseudoviruses were indistinguishable in terms of their locations and numbers of observed spokes (*Figure 3*; *Figure 3—figure supplement 1*; *Figure 3—source data 1*). We then gathered data for 6535-ΔCT attachment sites, finding a small, but statistically significant (p=$3\times10^{-4}$) increase in the mean number of spokes: 3.9 +/- 1.2 (*n* = 15) spokes for 6535-ΔCT as compared to 2.4 +/- 0.6 (*n* = 116) spokes for wild-type pseudoviruses (*Figures 3d* and *5*; *Figure 3—source data 1*).

## Discussion

Despite a wealth of information about the pre- and post-fusion structures of HIV-1 Env and Env's interactions with host receptors (*Ward and Wilson, 2017*; *Ozorowski et al., 2017*; *Wang et al., 2018*; *Wang et al., 2016*; *Yang et al., 2019*; *Shaik et al., 2019*; *Chan et al., 1997*; *Weissenhorn et al., 1997*; *Liu et al., 2008*), the architecture of the virus-host cell contact and the structure of Env during the act of fusion have remained elusive. Because HIV-1 virions have not been visualized during the act of fusion, we assumed that viral fusion is a fast process that has yet to be captured by electron microscopy imaging of either cultured cells or tissues. In order to visualize the hypothesized pre-hairpin intermediate structure in which the gp41 transmembrane region and the fusion peptide link the viral and target cell membranes (*Chan and Kim, 1998*), we incubated HIV-1 fusion inhibitors that bind to the N-trimer region of gp41 that is exposed upon Env binding to the host receptor and coreceptor on target cells. We chose to use diverse fusion inhibitors to trap the hypothesized pre-hairpin intermediate structure linking the viral and host cell membranes (*Figure 1a*) and to avoid a temperature jump protocol (*Sougrat et al., 2007*) that could result in unanticipated effects on membranes; for example, membrane trafficking events can be altered or stopped at decreased, non-physiological temperatures, and structures associated with them, such as Golgi cisternae, are perturbed and noticeably altered relative to physiological conditions (*Ladinsky et al., 2002*).

We chose three fusion inhibitors with strong, medium, and weak neutralization potencies (CPT31, T1249-Fc, D5 IgG, respectively), incubated them at 37 ˚C with virus and target cells for variable times, and examined them in 3-D by ET. For all experimental conditions, we found virions linked to the surface of target cells by several (mean = 2.4 +/- 0.5; *n* = 116) narrow spokes (*Figure 3d*), thus all conditions contained sufficient numbers of virions and inhibitors to form stable attachments to target cells. By contrast, in control experiments with either no inhibitor, an irrelevant protein, or inhibitor added at a concentration equal to 1% of its neutralization $IC_{50}$ value, we did not observe virions attached to target cells. Based on current understanding of how fusion inhibitors prevent fusion (*Chan and Kim, 1998*; *Eckert and Kim, 2001*; *Figure 1a,b*), we conclude that the attachment

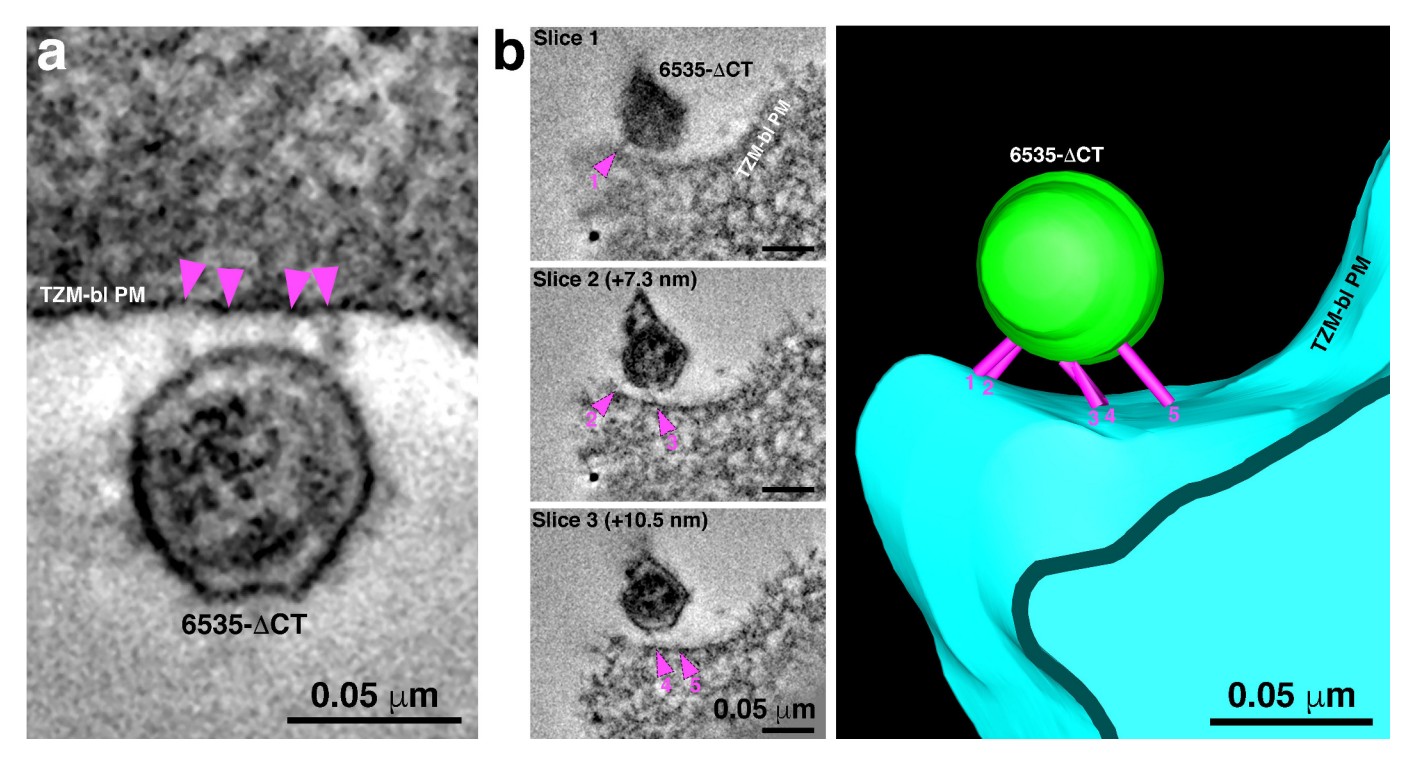

**Figure 5.** 6535ΔCT pseudoviruses are often attached to target cells with more than 2–3 spokes. (A) An example of a 6535ΔCT virion attached to the plasma membrane of a TZM-bl cell (cell and virus were treated with the T1249-Fc fusion inhibitor for 2 hr) by four distinct spokes (arrowheads). (B) An example shows a 6535ΔCT virion attached to the plasma membrane of a TZM-bl cell (cell and virus were treated with the T1249-Fc fusion inhibitor for 2 hr). Viewing the tomogram in a series of slices through the reconstruction (left panels) reveals five distinct spokes (magenta arrowheads) at different positions on the virus' surface. 3-D model of the reconstruction (right) displays the five spokes (magenta) attaching the virion (green) to the TZM-bl plasma membrane (PM) (blue).

densities represent Env trimers in the pre-hairpin intermediate conformation that link viral and target cell membranes (*Figure 1a*).

Having trapped virions in the act of fusion, we can use information gathered from tomograms to address mechanistic and structural details of fusion between viral and target cell membranes. First, the results suggest that only 2 to 3 (and occasionally 4) Envs participate in the reaction with target cells that is inhibited by a fusion inhibitor. This number, which could be somewhat inflated due to trapping of Envs using a fusion inhibitor, is consistent with studies suggesting that only a few HIV-1 Envs are required for fusion with a target cell membrane (*Yang et al., 2005*; *Magnus et al., 2009*). The number also represents a minority of the Env trimers on HIV-1, even considering that HIV-1 includes a unusually low number of spikes per virion (an average of ~14, with a range from 4 to 35 [*Zhu et al., 2003*; *Zhu et al., 2006*; *Liu et al., 2008*; *Chertova et al., 2002*; *Layne et al., 1992*], compared with ~450 spikes on similarly-sized influenza A virions [*Yamaguchi et al., 2008*]). STED microscopy studies suggested that HIV-1 spikes rearrange from a random distribution on immature virions to a cluster on mature virions (*Chojnacki et al., 2012*; *Chojnacki et al., 2017*; *Carravilla et al., 2019*). However, interspike distance distributions on mature virions derived from independent cryo-ET reconstructions revealed relatively random spike distributions rather than a single cluster of spikes (*Zhu et al., 2006*; *Liu et al., 2008*). Our finding of only 2–3 spokes per virion-target cell attachment site (*Figure 3d*; *Figure 3—source data 1*) is also consistent with random spike distributions on mature virions since more attachment spokes would be expected if Envs are clustered. In addition, although free Env trimers on HIV-1 virions are difficult to identify conclusively by ET using stained, plastic-embedded samples, we sometimes found Env spikes on the opposite surface of attached mature virions as the attachment structures (*Figure 4c,d*), consistent with the assumption that only two or three Envs are usually close enough to each other to participate in

binding to host cell receptors and that slow lateral diffusion of HIV-1 Envs within the virion bilayer prohibits recruitment of additional Envs into the attachment structure. Interestingly, membrane fusion in other systems, for example mediated by influenza hemagglutinin or SNARE complexes, also requires only ~three fusion proteins (*Ivanovic et al., 2013*; *Mohrmann et al., 2010*), suggesting mechanistic similarities that would be interesting to explore.

The ectodomains of Env trimers on immature virions are not expected to exhibit conformational changes that would prevent binding to CD4 and coreceptor. Indeed, immature virions can fuse with targets, although less efficiently than mature virions (*Wyma et al., 2004*; *Murakami et al., 2004*). Consistent with Env trimers on immature virions binding to receptors on target cells and forming pre-hairpin intermediate structures, we occasionally found immature virions attached to target cells that were incubated with a fusion inhibitor. Immature virions comprised <5% of the datasets recorded. Attached immature virions exhibited comparable numbers and locations of spokes as we found for attached mature virions (*Figure 4d*), consistent with a similar distribution of Envs on mature and immature virions. Although measurements of Env mobility within the membranes of mature and immature virions differ, the HIV-1 viral membrane on both mature and immature virions is a low mobility environment, likely due to high lipid order (*Chojnacki et al., 2017*). Our results suggest that the intrinsically low mobilities of HIV-1 Envs on both mature and immature virions limit the number of Env trimers that can participate in pre-hairpin intermediate structures with target cells.

The cytoplasmic tail of HIV-1 Env interacts directly with the viral matrix protein (*Alfadhli et al., 2019*), and tail deletion has been suggested to increase the lateral mobility of Envs in the viral membrane (*Crooks et al., 2008*; *Roy et al., 2013*; *Pezeshkian et al., 2019*). In order to determine the effects of cytoplasmic tail deletion on virion attachment to a target cell in the presence of a fusion inhibitor, we used ET to compare attachment sites for virions with wild-type and cytoplasmic tail-deleted Envs. We found a slight, but statistically significant, increase for the number of spokes in attachment sites formed between fusion inhibitor-treated target cells and the 6535-ΔCT versus wild-type pseudoviruses (3.9 +/- 1.2 for 6535-ΔCT to 2.4 +/- 0.5 for wild-type) (*Figure 3d*; *Figure 3— source data 1*). These results are consistent with intrinsically low mobilities of HIV-1 Envs in the viral membrane that are increased only slightly for Envs that lack a cytoplasmic tail. Taken together, analyses of the numbers of apparent pre-hairpin intermediate structures trapped by fusion inhibitors on wild-type mature, wild-type immature, and cytoplasmic tail-deleted virions is consistent with the assumption that a minority of the relatively few HIV-1 Env trimers are involved in attachment via a pre-hairpin intermediate to target cells.

We also used the tomographic data to measure the dimensions at the virion-target cell attachment sites, finding that the approximate dimensions of the majority of the spokes (15.6 nm ±2.8 nm in length x 3.9 ± 0.8 nm in width; n = 20) are consistent with a pre-hairpin intermediate structure formed by gp41 alone after gp120 has dissociated (step iv, *Figure 1a*), which we measured from an approximate model to be ~100 Å x ~ 22 Å (*Figure 1b*). By contrast, a hypothetical pre-hairpin intermediate containing gp120 (step iii, *Figure 1a*) would be wider, with dimensions ~ 100 Å by ~120 Å (*Figure 1b*). This wider structure is roughly consistent with the ~100 Å x ~ 100 Å dimensions of each of the 5–7 rods of density that comprised the 'entry claw' in a previous ET study (*Sougrat et al., 2007*). We suggest that the entry claw rods visualized after incubating virus-target cell samples at 4 ˚C and then warming to 37 ˚C represent a structure formed after the HIV-1 Env trimer has bound to CD4 (and possibly also to a coreceptor) but prior to gp120 dissociation. Given that entry claws were not visualized in the temperature jump protocol when the viruses and target cells were incubated in the presence of a fusion inhibitor (*Sougrat et al., 2007*), the thick rod structures are unlikely to represent a pre-hairpin intermediate. We therefore suggest that the thick rods seen following temperature jump likely correspond to a structure resembling step ii in *Figure 1a*.

A potential explanation for why we observe 2–3 densities at virion-target cell attachment sites versus the 5–7 reported previously (*Sougrat et al., 2007*) could relate to the observation that the attachment surfaces in our imaging experiments were generally flat, as opposed to including a concave surface on the target cell that followed the convex surface of the virion (*Sougrat et al., 2007*). This difference could result from use of different sample preparation protocols and/or target cells (T cells in *Sougrat et al., 2007* versus TZM-bl cells in our study). In any case, if HIV-1 virions contain a random assortment of a small number of Env trimers that diffuse slowly or not at all in the membrane, then only a small number of Envs would be available on the small contact surface formed by a roughly spherical virion and a flat cell membrane, whereas more Envs would be available for

contacts with a larger contact surface formed by a virion interacting with a concave cell membrane (*Figure 1c*). Indeed, a recent ET study of murine leukemia virus (MLV) attached to target cells revealed a concave surface on the target cell with ~28 spokes per attached virion (*Riedel et al., 2017*), a number that would be predicted to be higher than observed for HIV-1 even if HIV-1 were interacting with a concave portion of its target cell because MLV includes more spikes (at least 100 per virion) (*Stano et al., 2017*) compared with HIV-1 (7–14 per virion) (*Zhu et al., 2003*; *Zhu et al., 2006*; *Liu et al., 2008*; *Chertova et al., 2002*; *Layne et al., 1992*).

In summary, we have used fusion inhibitors and ET of optimally preserved samples to visualize HIV-1 virions caught in the act of fusion with target cell membranes. These experiments revealed details of attachment sites with spokes representing pre-hairpin intermediate structures of HIV-1 Env. The spokes likely correspond to extended Env trimers after gp120 dissociation and prior to collapse into the post-fusion six-helical bundle structure (step iv in *Figure 1a*). Our observation of relatively few (2-3) spokes per attached virion implies that HIV-1 Env-mediated membrane fusion may require only a fraction of the ~14 Envs on each virion (*Zhu et al., 2003*; *Zhu et al., 2006*; *Liu et al., 2008*; *Chertova et al., 2002*; *Layne et al., 1992*) and provides further details of an intermediate step on the pathway to viral entry.

# Materials and methods

## Preparation of fusion inhibitors

A gene encoding T1249-Fc was constructed to encode the 39-residue T1249 peptide sequence (*Eron et al., 2004*) fused at its C-terminus to a $(Gly_4Ser)_7$ linker followed by human IgG1 Fc. T1249-Fc were expressed by transient transfection in 293-6E (CNRC) or Expi293 (ThermoFisher Scientific) cells and purified from transfected cell supernatants using a HiTrap MabSelect SuRe column (GE Healthcare) followed by size exclusion chromatography (SEC) using a Superdex 200 column (GE Healthcare), and its concentration was determined by $A_{280\ nm}$ measurements using an extinction coefficient of 129,550 $M^{-1}\ cm^{-1}$. D5 IgG was obtained from the NIH AIDS Reagents program at a stock concentration of 8.5 mg/mL. CPT31 was synthesized as described with its concentration measured by $A_{280}$ (extinction coefficient of 37,980 $M^{-1}\ cm^{-1}$) (*Redman et al., 2018*). Z004 IgG, an anti-Zika virus antibody used as a control for non-specific entry inhibition, was expressed and purified as described (*Robbiani et al., 2017*).

## In vitro neutralization assays

SC4226618, 6535, and 6535-ΔCT (Env gene truncated after stop codon corresponding to gp41 residue Phe752) pseudoviruses were produced by cotransfection of HEK 293 T cells with an Env expression plasmid and a replication-defective backbone plasmid (*Montefiori, 2005*). In vitro neutralization assays were performed by measuring the reduction of HIV-1 Tat-induced luciferase reporter gene expression in the presence of a single round of pseudovirus infection in TZM-bl cells (*Montefiori, 2005*). Inhibitors were evaluated using a four-fold inhibitor dilution series (each concentration run in duplicate). Nonlinear regression analysis was used to derive $IC_{50}$ and $IC_{90}$ values, the concentrations at which half-maximal and 90% inhibition, respectively, were observed.

## Incubations for fusion inhibitor ET experiments

TZM-bl cells were plated and cultured as described (*Montefiori, 2005*) on carbon-coated, glow-discharged synthetic sapphire disks (3 mm diameter, 0.05 mm thickness; Technotrade International). Briefly, ~50,000 to 70,000 cells in 1 mL were seeded in each well and then replaced after a day with fresh media containing DEAE-Dextran (12 μg/mL). T1249-Fc, CPT31, or D5 IgG fusion inhibitor, each at a concentration of 130 μg/mL, was combined with pseudovirus (either SC4226618, 6535, or 6535-ΔCT, each at ~5000 $TCID_{50}$/mL), and then immediately added to the cells and incubated for 2, 4, or 48 hr at 37 ˚C. For control experiments, incubations were done with either no fusion inhibitor or with or with 130 μg/mL control IgG (Z004). In another control with SC4226618 pseudovirus, we added 0.01 μg/mL T1249-Fc fusion inhibitor (a concentration corresponding to 100-fold lower than the $IC_{50}$ value for T1249-Fc against SC4226618). For all incubations, supernatant was then removed from each well, and cells were lightly fixed with 3% glutaraldehyde, 1% paraformaldehyde, 5% sucrose in

0.1 M sodium cacodylate trihydrate to render the samples safe for use outside of BSL-2 containment.

## EM preparation and electron tomography

Sapphire disks were rinsed briefly with 1-Hexadecene (Sigma), placed individually into brass planchettes (Type A/B; Ted Pella, Inc), and rapidly frozen with a HPM-010 High Pressure Freezing machine (BalTec/ABRA). Disks were transferred under liquid nitrogen to cryo-vials (Nunc) containing 2.5% $OsO_4$, 0.05% uranyl acetate in acetone and then placed in a AFS-2 freeze substitution machine (Leica Microsystems, Vienna). Samples were freeze substituted at $-90°C$ for 72 hr, warmed to $-20°C$ over 12 hr, held at that temperature for 24 hr, and then warmed to room temperature and infiltrated with Epon-Araldite resin (Electron Microscopy Sciences, Port Washington, PA). Sapphire disks were flat-embedded on teflon-coated glass slides with Secure-Seal adhesive spacers (Sigma) and Thermanox plastic coverslips (Electron Microscopy Sciences). Resin was polymerized at 60°C for 24 hr.

Once embedded, the sapphire disks were removed, leaving the cells as a monolayer within the resin wafer. The cells were observed with an inverted phase-contrast microscope to ascertain preservation quality and to select regions of interest (i.e. regions with >10 cells in close proximity). These regions were extracted from the resin wafer with a microsurgical scalpel and glued to plastic sectioning stubs. Semi-thick (300–400 nm) serial sections were cut with a UC-6 ultramicrotome (Leica Microsystems, Vienna) using a diamond knife (Diatome, Ltd., Switzerland). Sections were collected onto formvar-coated copper-rhodium 1 mm slot EM grids (Electron Microscopy Sciences) and stained with uranyl acetate and lead citrate. Colloidal gold particles (10 nm) were placed on both surfaces of the grid to serve as fiducial markers for subsequent tomographic image alignment.

Grids were placed in a dual-axis tomography holder (Model 2040; E.A. Fischione Instruments, Export, PA) and imaged with a Tecnai TF-30ST transmission electron microscope (Thermo-Fisher Scientific) operating at 300 keV. Images were recorded with a XP1000 CCD camera (Gatan, Inc). For dual-axis tomography, grids were tilted +/- 64° and images taken at 1° intervals. The grid was then rotated 90° and a similar tilt-series was taken about the orthogonal axis. Tilt-series datasets were acquired automatically using the SerialEM software package (*Mastronarde, 2005*). Tomograms were calculated, analyzed and modeled (including isosurface renderings) using the IMOD software package (*Kremer et al., 1996*; *Mastronarde, 2008*; *Mastronarde and Held, 2017*) on Mac Pro and iMac Pro computers (Apple, Inc). Briefly, individual tomograms were prepared from the aligned tilt series based on models of the positions of ~50 fiducial markers in each tilted image, and then re-projected using an R-weighted back-projection algorithm (*Sandberg and Brega, 2007*). Individual tomograms were joined to form a single reconstruction with less information loss than a single-axis tomogram due to the smaller missing wedge.

## Identification and imaging of HIV-1 virions

Prior to collecting tomographic data, HIV-1 virions were identified as follows (*Figure 2*; *Figure 2—figure supplement 1*): Thick sections were observed in the electron microscope and peripheries of cells were surveyed at medium magnification (3900x – 6500x). Objects that appeared to be spherical, were estimated to have a diameter of ~100 nm, and were proximal to a cell surface were examined at higher magnification (12,000x – 15,000x). These objects were observed at 0° tilt (perpendicular to the beam) and at +35° and −35° tilts to confirm that they were indeed spherical. Nonspherical objects, such as thin cellular projections or microspikes, would appear oblong or tubular at one or both high-tilt views (*Figure 2—figure supplement 1c*). Objects that remained spherical were further evaluated by observing through a defocus series to detect core structures that would be indicative of a HIV-1 virion (i.e. a bullet-shaped mature core or a C-shaped immature core [*Benjamin et al., 2005*; *Wright et al., 2007*]). Detection of core structures allowed the object to be classified as a HIV-1 particle, and it was subsequently imaged for dual-axis tomography (*Figure 2—figure supplement 1c*). In most sample preparations, attached virions were found at an incidence of ~1 per every five cells in a given 400 nm section. On rare occasions, several (2-3) virions were found attached to a single cell, and each virion was recorded as a separate dataset. Spoke counts were determined from 3-D tomographic reconstructions (*Figure 3d*; *Figure 3—source data 1*). Significance evaluations of spoke count differences between 6535-ΔCT and wild-type pseudoviruses was performed using a two-sample *t*-test assuming unequal variances.

## Acknowledgements

The authors thank Annie Lynch for assistance with initial experiments, Erica Lee, Jennifer Keeffe, and the Beckman Institute Protein Expression Center at Caltech for preparing T1249-Fc, D5 IgG, and Z004 IgG, Sarah Apple and Nicholas Francis for preparing CPT31, Magnus Hoffmann for suggestions, and members of the Bjorkman and Kay laboratories for helpful discussions and critical review of the manuscript. This work was supported by the National Institute of General Medical Sciences (2 P50 AI150464) (to MSK and PJB). We thank the Caltech Kavli Nanoscience Institute for maintenance of the TF-30 electron microscope.

## Additional information

### Competing interests

Pamela J Bjorkman: Reviewing editor, *eLife*. The other authors declare that no competing interests exist.

### Funding

| Funder | Grant reference number | Author |
|---|---|---|
| National Institute of Allergy and Infectious Diseases | 2 P50 AI150464 | Michael S Kay Pamela J Bjorkman |

The funders had no role in study design, data collection and interpretation, or the decision to submit the work for publication.

### Author contributions

Mark S Ladinsky, Conceptualization, Data curation, Formal analysis, Supervision, Funding acquisition, Visualization, Methodology, Writing - original draft, Writing - review and editing; Priyanthi NP Gnanapragasam, Conceptualization, Resources, Data curation, Formal analysis, Visualization, Methodology, Writing - original draft, Writing - review and editing; Zhi Yang, Resources, Data curation, Methodology, Writing - review and editing; Anthony P West, Data curation, Formal analysis, Methodology; Michael S Kay, Conceptualization, Resources, Formal analysis, Methodology, Writing - review and editing; Pamela J Bjorkman, Conceptualization, Resources, Formal analysis, Supervision, Funding acquisition, Writing - original draft, Writing - review and editing

### Author ORCIDs

Mark S Ladinsky (iD) https://orcid.org/0000-0002-1036-3513
Michael S Kay (iD) http://orcid.org/0000-0003-3186-9684
Pamela J Bjorkman (iD) https://orcid.org/0000-0002-2277-3990

### Decision letter and Author response

Decision letter https://doi.org/10.7554/eLife.58411.sa1
Author response https://doi.org/10.7554/eLife.58411.sa2

## Additional files

### Supplementary files

• Transparent reporting form

### Data availability

Raw datasets are freely available upon request. Interested parties should contact ladinsky@caltech.edu, and we will place requested datasets onto an externally accessible Caltech Box Server. Requestors will then be provided with a direct URL link from which they can download the files at their convenience.

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
