## [Decision Letter]

**Acceptance summary:**

Ladinsky et al. report electron tomographic studies of HIV pseudovirions bound to TZM-bl cells, showing direct evidence for the elongated, "prehairpin" intermediate bridging virion and cell surface. They detect a substantial number of virions attached to the cell by between 2 and 4 thin "spokes", about 130 Å long and spaced by between 70 and 140 Å. These images suggest that the spokes correspond to the extended pre-fusion state of HIV gp41 fusion protein.

**Decision letter after peer review:**

Thank you for submitting your article "Electron tomography visualization of HIV-1 fusion with target cells using inhibitors to trap the prehairpin intermediate" for consideration by *eLife*. Your article has been reviewed by three peer reviewers, one of whom is a member of our Board of Reviewing Editors, and the evaluation has been overseen by John Kuriyan as the Senior Editor. The following individuals involved in review of your submission have agreed to reveal their identity: Stephen C Harrison (Reviewer #2), Eric Freed (Reviewer #3).

The reviewers have discussed the reviews with one another and the Reviewing Editor has drafted this decision to help you prepare a revised submission.

Summary:

Ladinsky et al. report electron tomographic studies of HIV pseudovirions bound to TZM-bl cells, showing direct evidence for the elongated, "prehairpin" intermediate bridging virion and cell surface. They have done a careful job of setting the experimental conditions to avoid temperature jump, well known to produce artifacts in membrane traffic both at the plasma membrane and in the Golgi, by trapping the intermediate with each of three different fusion inhibitors that bind the three-helix coiled-coil formed by the segment often called HR1. They have also (correctly) avoided prolonged fixation by using freeze-substitution to achieve an effective "freeze frame". Electron tomograms of 300-400 nm negatively stained sections of high-pressure frozen / freeze-substitution fixed samples were collected. They detect a substantial number of virions attached to the cell by between 2 and 4 thin "spokes", about 130 Å long and spaced by between 70 and 140 Å. They can find no such particles when they do not trap with an inhibitor consistent with studies on other viruses showing that when an appropriate number of extended intermediates have formed adjacent to one another, bridging the two membranes, fusion ensues very rapidly. These images suggest that the spokes correspond to the extended pre-fusion state of HIV gp41 fusion protein.

Essential revisions:

1) Approximate dimensions of the spokes are provided in the sixth paragraph of the Discussion (…majority… of the spokes, 100-150 Å in length x 20-35 Å in width). How were these dimensions measured considering that single tomograms are very noisy and suffer from missing wedge artifacts? Please provide statistics on these measurements.

2) A potential concern relates to the use of fusion inhibitors to stabilize the intermediate pre-fusion state. Could they affect the number of connections that the virus normally has to make to fuse with the host cell. Are there other data to support the number of observed connections per virus-host interactions? We note that an alternative would have been to use fusion-defective Env mutants, which would not have added density to the target structure (note, we are not requesting such an experiment). In any case, the authors might strengthen their case by discussing previously published results that 2-3 fusion proteins also appear to be enough for fusion of influenza virus with a target membrane and a similarly small number for SNARE-mediated fusion.

3) The cells were "lightly" fixed (with 3% glutaraldehyde, 1% paraformaldehyde, mentioned in the subsection “Incubations for fusion inhibitor ET experiments”) prior to high-pressure freezing. Why was this done? Were images collected without the fixing step? Could the fixing step have affected the quality of the images?

4) A number of studies have shown that HIV-1 VLPs, even in the absence of Env or CD4, bind to HeLa cells. It is therefore surprising that the authors observed no attached virions in the absence of fusion inhibitor. Please comment.

5) A few more connecting spokes are present with mutant virions that lacked the Env cytoplasmic tail (Figure 5). The images in slices 1-3 show deformed virus particles. Are there better examples of bound virions (in contrast, most of the wildtype virons shown in Figure 3—figure supplement 1 and in Video 1 are much "nicer").

6) There are clear differences in spike structures reported here and by Subramaniam (the "entry claw"). The authors ascribe these differences to the low temperature incubation performed in the Subramaniam study (Sougrat et al.). Another key difference is that the current authors used HeLa (TZM-bl) cells as targets whereas Sougrat et al. used T cells. The possibility that this difference in target cell selection could explain some of the differences should be discussed.

---

## [Author Response]

Essential revisions:1) Approximate dimensions of the spokes are provided in the sixth paragraph of the Discussion (…majority… of the spokes, 100-150 Å in length x 20-35 Å in width). How were these dimensions measured considering that single tomograms are very noisy and suffer from missing wedge artifacts? Please provide statistics on these measurements.

The missing wedge was minimized by preparing tomograms from dual-axis tilt-series. Briefly, the target was imaged at 1° intervals through a +/-64° series. The sample was then rotated 90° within the microscope and a second tilt-series with identical parameters was taken about the orthogonal axis. The two tomograms were then merged to make a single reconstruction with a 50% smaller missing wedge, that preserves the inherent resolution of the data throughout its volume (see Mastronarde, D.N. (1997) Dual-axis tomography: An approach with alignment methods that preserve resolution. J. Struc. Biol. 120: 434-352). When necessary, tomograms were noise-filtered during processing and/or median-filtered during postprocessing to confirm the nature of spokes. Spokes were measured by viewing each one at an optimum tomographic angle (i.e., where its complete length could be unambiguously viewed from the virion surface to the cell surface, and then modeled as a single line. The line was measured in terms of pixels and then transposed to Ångstroms based on the pixel size of the original data.

In response to the request for statistics on these measurements, we now report the length and width of spokes as 15.6 ± 2.8 nm and 3.9 ± 0.8 nm (n = 20) in the text of the revised paper.

2) A potential concern relates to the use of fusion inhibitors to stabilize the intermediate pre-fusion state. Could they affect the number of connections that the virus normally has to make to fuse with the host cell. Are there other data to support the number of observed connections per virus-host interactions? We note that an alternative would have been to use fusion-defective Env mutants, which would not have added density to the target structure (note, we are not requesting such an experiment). In any case, the authors might strengthen their case by discussing previously published results that 2-3 fusion proteins also appear to be enough for fusion of influenza virus with a target membrane and a similarly small number for SNARE-mediated fusion.

The revised Discussion points out that if adding fusion inhibitors affects the number of connections the virus would use to fuse with the host cell, it would likely increase the number of connections. We don’t know how to definitively address this issue, however, since we can’t see the connections in the absence of an added fusion inhibitor. However, as pointed out by the reviewers, the number of connections we see correlates with findings in other fusion systems. In the revised paper (Discussion, third paragraph), we cited previous results showing that only a few influenza hemagglutinins and SNARES have been shown to be required for fusion.

3) The cells were "lightly" fixed (with 3% glutaraldehyde, 1% paraformaldehyde, mentioned in the subsection “Incubations for fusion inhibitor ET experiments”) prior to high-pressure freezing. Why was this done? Were images collected without the fixing step? Could the fixing step have affected the quality of the images?

As stated in the Materials and methods section, “…cells were lightly fixed with 3% glutaraldehyde, 1% paraformaldehyde, 5% sucrose in 0.1 M sodium cacodylate trihydrate to render the samples safe for use outside of BSL-2 containment.” The primary reason that samples were lightly fixed with aldehydes prior to high-pressure freezing was to render them safe for handling outside of BSL-2 containment, as mandated by Caltech safety protocols for handling of pseudovirus. Our Baltec HPM010 high-pressure freezer and ancillary equipment is not housed in a BSL-2 facility, nor is it portable. Indeed, chemical fixation can affect image quality, but the major source of artifact is introduced by room temperature dehydration. This step is eliminated by high-pressure freezing, and subsequent freeze-substitution, thus yielding samples that are ~80% more reliably preserved than traditional chemical fixation alone. Furthermore, controls conducted throughout the study did not show any attached virions, nor structures resembling spokes associated with free virions, indicating that these structures were not aldehyde cross-links or artifacts thereof.

4) A number of studies have shown that HIV-1 VLPs, even in the absence of Env or CD4, bind to HeLa cells. It is therefore surprising that the authors observed no attached virions in the absence of fusion inhibitor. Please comment.

We haven’t been able to locate papers discussing HIV-1 VLPs that bind to HeLa cells in the absence of Env or CD4. In any case, we did not find attached virions in control experiments in which we either did not add a fusion inhibitor or we added a control protein.

5) A few more connecting spokes are present with mutant virions that lacked the Env cytoplasmic tail (Figure 5). The images in slices 1-3 show deformed virus particles. Are there better examples of bound virions (in contrast, most of the wildtype virons shown in Figure 3—figure supplement 1 and in Video 1 are much "nicer").

We have added a new panel to Figure 5 that includes a clearer image of a 6535∆CT virion attached to a TZM-bl cell by 4 distinct spokes seen in a single tomographic view. The previously-used example remains as panel B, to show that spokes are not always found in a single tomographic plane and to display the bound virion as a 3D reconstruction. It was often the case that tomographic slices that clearly identified attachment spokes did not show the virion itself in an optimum orientation.

6) There are clear differences in spike structures reported here and by Subramaniam (the "entry claw"). The authors ascribe these differences to the low temperature incubation performed in the Subramaniam study (Sougrat et al.). Another key difference is that the current authors used HeLa (TZM-bl) cells as targets whereas Sougrat et al. used T cells. The possibility that this difference in target cell selection could explain some of the differences should be discussed.

The reviewer raises an excellent point about using different target cells, which we now address in the Discussion of the revised paper.